# Cannabis Hyperemesis Syndrome in Youth: Clinical Insights and Public Health Implications

**DOI:** 10.3390/ijerph22040633

**Published:** 2025-04-17

**Authors:** Jamie A. Seabrook, Morgan Seabrook, Jason A. Gilliland

**Affiliations:** 1Department of Epidemiology & Biostatistics, Western University, London, ON N6G 2M1, Canada; jgillila@uwo.ca; 2Department of Paediatrics, Western University, London, ON N6A 5W9, Canada; 3Brescia School of Food and Nutritional Sciences, Western University, London, ON N6G 2V4, Canada; 4Children’s Health Research Institute, London, ON N6C 2V5, Canada; 5Lawson Research Institute, London, ON N6A 4V2, Canada; 6London Health Sciences Centre Research Institute, London, ON N6A 5W9, Canada; 7Human Environments Analysis Laboratory, Western University, London, ON N6A 3K7, Canada; mseabro5@uwo.ca; 8Department of Geography and Environment, Western University, London, ON N6A 5C2, Canada; 9School of Health Studies, Western University, London, ON N6A 3K7, Canada

**Keywords:** cannabis hyperemesis syndrome, youth, cannabis use, vomiting, dehydration, mental health, treatment, public health, abdominal pain, intervention

## Abstract

This review focuses on Cannabis Hyperemesis Syndrome (CHS) in youth, a condition linked to chronic cannabis use and characterized by cyclic vomiting, abdominal pain, and dehydration. The objectives were to explore CHS progression in youth and its impact on health, and to assess current treatment strategies. There are the three distinct phases of CHS: prodromal, hyperemetic, and recovery. During the prodromal phase, individuals experience early morning nausea and discomfort, often mistakenly alleviated by continued cannabis use. The hyperemetic phase is marked by severe vomiting, dehydration, and complications like electrolyte imbalances, leading to potentially serious health risks. Temporary relief may be experienced through hot showers or baths. In the recovery phase, symptoms gradually resolve, and normal eating and bathing habits return. The review emphasizes the physical and psychological impacts of CHS on youth, highlighting the potential for misdiagnosis and the importance of early intervention. It stresses the need for targeted educational efforts in schools, healthcare settings, and public health campaigns to prevent delayed diagnosis and improve outcomes. Findings underscore the importance of increasing healthcare provider awareness and promoting preventive education. The review also advocates for further research into CHS pathophysiology to improve diagnostic and treatment protocols for young populations.

## 1. Introduction

Youth cannabis use is a growing global health issue. In 2021, the global annual prevalence of cannabis use among youth aged 15–16 years was 5.3%, with Europe reporting a higher rate of 11.0% [1]. Meanwhile, in the United States, cannabis use among 19–22-year-olds reached a historic high, with 42.6% reporting use in the past year [2]. Canada also ranks among the highest globally for youth cannabis use [3]. According to the 2024 Canadian Cannabis Survey, 41% of Canadians aged 16–19 reported using cannabis in the past year, with 20% reporting use within the past month and 9% engaging in daily or almost daily use. Usage rates peaked among Canadians aged 20–24, with 48% reporting cannabis use in the past year [4].

A significant concern with youth cannabis use is the increasing potency of delta-9-tetrahydrocannabinol (THC), the primary psychoactive compound responsible for the “high” associated with cannabis consumption [5,6,7,8]. THC potency has risen by 400% over the past four decades, increasing from 3% in the 1980s to 15% in 2023, with some strains averaging as high as 30% [9]. Since the brain continues to develop until approximately age 25, THC exposure during this critical period can disrupt neural development, impairing the formation and pruning of neural connections. This disruption can result in long-term cognitive deficits, including difficulties with attention, memory, and learning [10]. Youth cannabis use is also linked to an elevated risk of depression and anxiety [11,12], psychosis and paranoia [13], self-harm [14], and a heightened likelihood of developing cannabis use disorders [15].

A growing concern associated with high-potency cannabis is Cannabis Hyperemesis Syndrome (CHS), a gastrointestinal condition that affects individuals who use cannabis regularly (e.g., daily to weekly) over an extended period (from several months to years). CHS is characterized by severe and persistent vomiting lasting for hours or days, chronic nausea that worsens with cannabis use, abdominal cramping, and temporary symptom relief from hot showers or baths [5,6,8,16,17,18,19,20,21,22,23,24,25,26]. While definitive causation has not been established, emerging evidence suggests that chronic use of high-potency cannabis may be associated with a heightened risk of developing CHS [8]. Younger individuals are also disproportionately affected compared to middle-aged or older adults [6]. CHS was not documented in the literature until 2004 [27], likely due to the limited clinical recognition of the syndrome and a lack of formal case documentation. As a result, many clinicians and cannabis users remain unaware of the condition [28], highlighting the need for increased education and awareness.

A systematic review of 21 studies on CHS diagnosis and management in adolescents, involving a total of 24 patients, revealed that females were more likely than males to present with CHS, and 21% of the patients had a history of anxiety or depression [29]. Although large-scale studies are limited, recent research from Ontario, Canada, highlighted a significant increase in CHS-related emergency department visits among individuals aged 19–24 after the introduction of a legal commercial cannabis market (March 2020–June 2021), with an incident rate ratio of 1.60 (95% confidence interval, 1.19–2.16) [30]. The researchers argued that the commercialization of cannabis markets may have contributed to the rising rates of CHS-related emergency visits [30], a concern that is frequently echoed by others [6,16,21,23,26,31,32].

Given the limited awareness of CHS among both cannabis users and healthcare providers [5,6,26,28,33], this article provides a comprehensive and up-to-date overview of CHS in youth. Building on recent advancements in the literature, it synthesizes the latest evidence on CHS symptoms, diagnostic challenges, and phases of the condition, while also examining its physical and mental health impacts. This review goes further by addressing nutritional considerations, potential misdiagnoses of eating disorders, and the implications of cannabis legalization on youth health. With cannabis use becoming more prevalent and socially accepted, especially among young people, it is critical to raise awareness of CHS and support early intervention, prevention strategies, and clinical education. We hope this expanded scope and recency of evidence will make the paper a useful reference for both clinicians and public health professionals.

## 2. Materials and Methods

This article is a narrative review that synthesizes the most recent and relevant literature on CHS in youth. The review was guided by a set of predefined thematic areas, including symptomatology and diagnosis, symptom phases, nutritional considerations, physical and mental health impacts, and implications for prevention, education, and public health. These themes were developed in advance and used to guide both the literature search and synthesis process.

An iterative literature search was conducted from October 2024 to March 2025 using PubMed and Google Scholar. Search terms included combinations of keywords such as “Cannabis Hyperemesis Syndrome”, “CHS”, “cyclic vomiting” “adolescents”, “youth”, “cannabis use”, “vomiting”, “prevalence”, “pathophysiology”, “nutrition”, and “treatment”. Additional studies were identified by screening the reference lists of key papers. No restrictions were placed on language, publication date, or geographic location.

Peer-reviewed articles were included if they focused on CHS and addressed at least one of the predefined themes relevant to youth. Eligible studies covered the clinical presentation, diagnosis, symptom progression, treatment and management strategies, prevalence, or public health implications of CHS in adolescents or young adults. Only original research articles were included, such as randomized controlled trials, observational studies, case reports, case series, and qualitative studies. Review articles, expert commentaries, and opinion pieces were excluded to avoid the duplication of findings and maintain focus on primary data sources. Studies were also excluded if they did not focus specifically on CHS or lacked relevance to youth populations.

Articles were screened manually, and relevant data were extracted and synthesized according to thematic relevance. Table 1 provides an overview of the 13 articles included in this review, spanning publications from 2004 to 2024. These articles consisted of 9 case-based studies (6 case reports and 3 case series), 1 randomized controlled trial, 1 longitudinal analysis, 1 repeated cross-sectional study, and 1 qualitative study. Every effort was made to include the most up-to-date and comprehensive literature available up to March 2025.

A total of 41 peer-reviewed articles were identified through this process. Titles and abstracts were manually screened to assess relevance to CHS in youth, with three articles excluded for lack of relevance. Full-text review was conducted for the remaining 38 articles. Of these, 25 were excluded because they did not present original research data (e.g., narrative reviews, systematic or scoping reviews, expert commentaries). Thirteen original research articles met inclusion criteria and were included in the final synthesis. A modified PRISMA flow diagram is provided in Figure 1 to illustrate this process.

## 3. Symptoms and Diagnosis of Cannabis Hyperemesis Syndrome

CHS is a condition that manifests in individuals who use cannabis frequently over a prolonged period. In their pioneering study, Allen et al. [27] reported on 10 patients from South Australia who experienced cyclical vomiting attributed to chronic cannabis use. The authors noted that 9 out of 10 patients sought relief through multiple hot showers or baths during the acute phase of their illness, and symptoms resolved with cannabis abstinence. While these observations provided an important initial link between chronic cannabis use and cyclical vomiting, the small sample size limited the generalizability of their findings. Additionally, the lack of a control group or randomization underscored the need for larger, more rigorously controlled studies to confirm causality [42].

Since this initial report, cases of CHS have been documented worldwide [6,26], with the first reports emerging from Germany in 2011 [37] and Puerto Rico in 2015 [36]. Emergency department visits for CHS in Canada and the United States have also doubled between 2017 and 2021 [16], highlighting the growing prevalence of this condition as cannabis use becomes more widespread.

The pathophysiology of CHS is complex and is only recently becoming more understood. Paradoxically, while low doses of cannabis can reduce nausea, chronic or high-doses can lead to CHS [11,19]. This is thought to result from disruptions in the endocannabinoid system, the pituitary-adrenal axis, and sympathetic nervous system regulation, with genetic predispositions and stress often playing significant roles [11]. Adolescents are known to experience higher levels of stress than older individuals, a developmental reality that can interact with genetic predispositions to increase the likelihood of adverse health outcomes [43]. Chronic cannabis use may also impair gastric emptying and contribute to symptoms through THC accumulation in fat cells, illustrating the dual anti-emetic and pro-emetic effects of cannabis [11].

A hallmark symptom of CHS is cyclic vomiting, characterized by episodes of severe nausea and vomiting separated by asymptomatic periods lasting weeks or months [6,8,16,18,26,44]. These episodes often include debilitating abdominal pain, and excessive vomiting can lead to dehydration, electrolyte imbalances, and disorientation [6,8,11,16,20,38,44]. A distinctive feature of CHS is the compulsive need for hot showers or baths to relieve symptoms [6,8,16,19,22,23,25,26,32,39]. Compulsive hot bathing is not an anxiety-related behavior but rather a learned response to symptom relief [6,8,26,27,32]. The most effective long-term treatment for CHS, however, is cannabis cessation [5,6,8,11,16,19,26,45]. A systematic review of case reports found that 96.8% of patients who stopped using cannabis experienced complete resolution of symptoms [8]. Symptoms typically resolve within days of stopping cannabis use [6,8], although they can reemerge within 24 h after the last use [16].

What remains unknown about CHS is why some chronic cannabis users develop the condition while others do not, and why symptoms manifest earlier in some individuals than in others [6]. This gap in understanding underscores the need for further research into the condition’s underlying mechanisms and risk factors. However, CHS is now diagnosed under the Rome IV criteria, which includes prolonged and excessive cannabis use, cyclic vomiting, resolution of vomiting episodes following sustained cannabis abstinence, and often, compulsive hot baths or showers for symptom relief [6,21,23,26,46]. This suggests that the diagnosis of CHS relies heavily on recognizing specific symptoms and patterns of patient behavior [6].

A systematic review of 211 CHS cases revealed a typical progression of the condition: the median age of first cannabis use was 16 years, symptoms began at 24 years, and diagnosis was made at 28 years. Among these patients, 71.6% reported daily cannabis use, while only 2.4% used cannabis less than once per week [8]. Importantly, there is no evidence that a single use or experimentation with cannabis can trigger CHS [6].

Despite being considered a rare condition by both clinicians and cannabis users, many cases of CHS remain undiagnosed or misdiagnosed. This is primarily because the symptoms—nausea, abdominal pain, and vomiting—overlap with numerous other medical conditions (e.g., gastroenteritis, peptic ulcer disease, cyclic vomiting syndrome), complicating the diagnostic process [6,8,35]. Addressing this gap in recognition and treatment is critical to improving outcomes for those affected.

While CHS is often underdiagnosed in adults, it is even more frequently overlooked in youth [6,8,18,38]. This is likely due to underreporting of cannabis use and other substances, driven by fear of judgment or stigma [47]. However, research indicates higher rates of CHS among individuals who use other substances [16]. For instance, cannabis use in youth is strongly linked to vaping [48,49], cigarette smoking [48,50], and alcohol consumption [50]. This lack of recognition by clinicians [5,6,28,33] may leave young people unaware that their symptoms are connected to chronic cannabis use. In some cases, youth may mistakenly believe that cannabis can relieve their cyclic vomiting, potentially exacerbating the condition [8].

Diagnosing CHS can thus be a lengthy and expensive process, often involving unnecessary procedures such as computed tomography (CT) scans, magnetic resonance imaging (MRI), gastric emptying tests, and colonoscopies [5,6,8,20,51]. Given these challenges, all youth presenting with cyclic vomiting should be systematically screened for cannabis use and compulsive hot bathing behaviors [8]. For patients who continue to experience CHS despite being informed of the link between their symptoms and cannabis use, referral to addiction specialists, substance use counselors, or rehabilitation programs may be necessary [5,6,45]. This might also suggest that cannabis is more addictive than previously understood.

## 4. Phases of Cannabis Hyperemesis Syndrome

The symptoms of CHS typically progress through three distinct phases: prodromal, hyperemetic, and recovery. During the prodromal phase, which can last for several months, individuals often experience early morning nausea, fear of vomiting, and abdominal discomfort. Despite these symptoms, they usually continue using cannabis, mistakenly believing it alleviates their nausea [21,26]. They also tend to maintain their usual dietary habits and body weight [21].

The hyperemetic phase is characterized by severe, persistent cyclic vomiting and intense abdominal pain, typically lasting 1–2 days, though it can sometimes extend longer [21,26,52]. This acute phase can lead to dehydration and electrolyte imbalances. Additional symptoms may include hot flashes, sweating, trembling, hypertension [21], weight loss, and difficulty keeping food down without vomiting [26]. It is often during this phase that individuals discover temporary relief from symptoms through very hot showers or baths. However, symptoms quickly return as the water cools or when they leave the hot water [26,27,53].

In the recovery phase, CHS symptoms fully resolve, marked by a return to normal bathing habits, regular eating patterns, and weight gain [21,26]. This phase typically begins within days to weeks of cannabis cessation [21]. One of the main challenges, however, is maintaining cannabis abstinence; symptom recurrence is common in individuals who resume cannabis use, which can perpetuate the cycle of illness [21,26]. While many patients experience complete symptom resolution with sustained abstinence, others may require behavioral support or addiction counseling to prevent relapse [21].

## 5. The Impact of Cannabis Hyperemesis Syndrome on Physical and Mental Health

Chronic cannabis use can have significant and sometimes severe physical and psychological effects on individuals. While the physical symptoms of CHS can be challenging, the condition also takes a toll on mental health, often intensifying emotional distress.

Frequent vomiting due to CHS may cause the erosion of tooth enamel, potentially leading to tooth loss [16]. Additionally, long-term chronic cannabis use and CHS can result in unintended weight loss, as individuals with the condition often struggle to retain food [6,18,26,38,44]. The condition can also cause severe dehydration and acute kidney injury, sometimes requiring emergency department visits and hospitalization [6,8,16]. In fact, it is common for CHS patients to present frequently to the emergency department and be hospitalized multiple times each year [6]. In rare instances, CHS can lead to death due to electrolyte imbalances resulting from chronic vomiting [6,22,26].

CHS can also negatively affect mental health. The emotional strain caused by frequent vomiting and dehydration can lead to increased anxiety, isolation, and depression, particularly when individuals are unaware that their chronic cannabis use is the underlying cause [21]. Anxiety and depressive symptoms may worsen when healthcare providers are unaware of the cause of the symptoms, especially during repeated visits to the emergency department or long hospital stays without a clear diagnosis. However, more research is needed to determine whether frequent emergency department visits for cannabis-related issues in youth contribute to greater mental health challenges [54].

Importantly, it is possible that some individuals began using cannabis in the first place to cope with pre-existing mental health conditions such as anxiety or depression [3]. In these cases, CHS symptoms may exacerbate, rather than cause, these underlying conditions. This distinction highlights the complex interplay between mental health and cannabis use, where cannabis may both contribute to and be used in response to psychological distress. Additionally, the cognitive impairment linked to CHS [10] could be aggravated by cyclic vomiting and dehydration, as well as the stress of repeated medical interventions.

The physical and psychological impacts of CHS may lead to an increase in substance use. Cannabis may be used as a coping mechanism, despite its potential to worsen CHS symptoms. In a qualitative study of semi-structured interviews with clinicians from California, one clinician noted about adolescents, “I’m seeing more and more who are using daily and who say that they have to in order to cope” [41]. Finally, the stigma surrounding CHS, coupled with the fear of being judged by family, friends, or healthcare providers, may discourage individuals from seeking treatment. This can result in heightened stress and increased feelings of isolation [46].

Furthermore, the negative effects of chronic cannabis use—such as cognitive dysfunction, motivational deficits, and poor academic performance—may intersect with the effects of CHS, compounding functional impairments in youth. For example, both CHS and long-term cannabis use may independently contribute to difficulties with attention, memory, and emotional regulation, potentially making it harder for affected individuals to recover or adhere to treatment recommendations [10,12,29,46].

## 6. The Importance of Early Intervention and Preventive Education

While there is no overarching acute or long-term evidence-based treatment protocol for CHS [11,26], the San Diego Emergency Medicine Oversight Commission published treatment guidelines in 2018 that recommended rehydration and supportive patient education and counsel to stop the use of cannabis [20]. More recently, the 2024 American Gastroenterology Association Clinical Practice Update recommended a combination of evidence-based psychosocial interventions and pharmacological treatments for management of CHS, although they did not discuss the efficacy of the acute treatments [11]. Regardless, early recognition of CHS is crucial to prevent complications arising from severe dehydration and fluid imbalance [23].

### 6.1. The Need for Youth-Specific Education

Education on CHS should begin before symptoms develop, particularly in youth who are more likely to use cannabis regularly while lacking awareness of its potential harms. Many CHS patients do not initially believe that cannabis can cause hyperemesis [6], making targeted education efforts in schools, healthcare settings, and public health campaigns essential for prevention. Misconceptions surrounding cannabis use—such as the belief that it is entirely safe or non-addictive—can delay diagnosis and treatment, leading to repeated emergency department visits and unnecessary medical testing [45].

### 6.2. Early Recognition and Treatment

Patients who go to the emergency department with dehydration caused by CHS usually receive intravenous fluids and antinausea medications [8,16,26,32]. However, conventional antiemetic therapy (e.g., ondansetron) offers little to no relief of symptoms [6,11,19,20,25,26,32]. Some evidence suggests that topical capsaicin (0.1%) cream may decrease nausea when applied to the upper abdomen [5,6,8,11,16,20,24,32,34]. Additionally, acute and short-term use of benzodiazepines (antianxiety medication) and haloperidol (antipsychotic medication) can be useful for some patients [5,8,11,16,20,32], while opioids should be avoided, as they may worsen symptoms and lead to dependence [8,20].

A randomized, triple-blind crossover trial from Ontario, Canada found that a one-time low dose of haloperidol at 0.05 or 0.1 mg/kg (n = 13 subjects) was superior to ondansetron 8 mg (i.e., n = 17 subjects) as a first-line treatment for acute CHS by reducing nausea and abdominal pain, decreasing the need for rescue antiemetics, and allowing earlier discharge from the emergency department [40]. Hospital admission may also resolve CHS by ensuring cannabis cessation during the inpatient stay [6]. However, recommending immediate cessation of cannabis can lead to withdrawal symptoms and a high risk of relapse [16,22,24], which should be carefully discussed with patients.

### 6.3. The Role of Primary Care and Public Health

Preventative education about CHS should be integrated into discussions about cannabis use in primary care, youth health visits, and school-based health programs. Given that youth may be resistant to quitting cannabis, healthcare providers should approach cessation discussions with harm reduction strategies, including gradual reduction plans, mental health support, and alternative coping mechanisms for those who use cannabis for stress relief or anxiety. Recognizing a patient’s resistance to stopping cannabis even temporarily can also serve as a clinical indicator of chronic cannabis use [6].

### 6.4. Healthcare and Societal Benefits of Early Intervention

Not only does early intervention spare the individual from unnecessary tests and procedures, but it also saves the healthcare system considerable time and resources [6]. Providing a definitive diagnosis at an early stage may increase the likelihood that young patients will acknowledge the role of cannabis in their symptoms and consider cessation [33,45]. A public health approach that combines early recognition, targeted education, and integrated treatment strategies has the potential to significantly reduce the burden of CHS on both patients and the healthcare system.

By fostering awareness among youth, parents, educators, and healthcare providers, CHS-related emergency visits can be minimized, and individuals struggling with cannabis-related health issues can receive timely and appropriate care.

### 6.5. Enhancing Disclosure of Cannabis Use in Youth

A significant barrier to the diagnosis and treatment of CHS in youth is the reluctance to disclose cannabis use to healthcare providers, particularly in emergency or clinical settings. This reluctance is often rooted in stigma surrounding cannabis use, fear of judgment, or concerns about legal repercussions [41]. To facilitate disclosure and ensure appropriate care, it is crucial for healthcare providers to create a supportive and non-stigmatizing environment.

One evidence-based recommendation is for providers to receive training on how to approach sensitive topics such as drug use in a non-judgmental manner. This can be achieved through the use of open-ended, neutral questions that avoid accusatory or stigmatizing language. For instance, asking “Have you been using cannabis or other substances recently?” rather than phrasing the question in a way that implies blame or judgment, such as “How much cannabis have you been using?” can help make patients feel more comfortable. This approach has been supported in research on youth substance use [3,10].

Further, medical providers should be educated on the importance of building trust with their patients. When youth feel that their healthcare provider is genuinely concerned about their well-being, they may be more likely to disclose their cannabis use, which in turn aids in the accurate diagnosis and effective treatment of CHS. Building trust and fostering an open dialogue is essential in addressing concerns about cannabis use among youth [12,41].

By fostering an open, empathetic dialogue, providers can not only increase the likelihood of cannabis disclosure but also encourage early intervention and prevention strategies for CHS, ultimately improving both short- and long-term outcomes for young patients [5,29].

## 7. Nutritional Considerations with Cannabis Hyperemesis Syndrome

When patients are hospitalized with CHS during the hyperemesis phase, treatment typically involves withholding all food and beverages, administering intravenous fluids to manage hydration, and gradually reintroducing clear liquids and food as symptoms improve and tolerance allows [11,24]. For those treated at home, fluids with glucose and electrolytes are recommended between vomiting episodes to stay hydrated [11].

As symptoms improve, patients may benefit from reintroducing bland, easily digestible foods, such as bananas, toast, rice, and broth, while avoiding fatty, spicy, or acidic foods that may trigger nausea. Due to the risk of prolonged vomiting, healthcare providers should monitor for electrolyte imbalances, including hypokalemia (low potassium) and magnesium deficiencies, which may require supplementation [11].

It is possible that CHS is misdiagnosed as an eating disorder because hyperemesis can become mistaken for a self-induced purging behavior [21,26]. Bulimia nervosa, for example, is a common eating disorder among young people and occurs in individuals who have difficulty controlling the quantity of food consumed (i.e., typically consuming more food in a 2 h period than most individuals), followed by purging behaviors [55,56]. This purging behavior after eating a large quantity of food may lead to a false diagnosis of bulimia nervosa when in fact it is really CHS [6]. However, a key distinction is that CHS-related vomiting is involuntary and not motivated by body image concerns, whereas bulimia involves intentional purging [6]. Moreover, about 35% of individuals who have substance use disorder also have an eating disorder [6]. Symptoms such as anorexia, dry heaving, headaches, and stomach pain are characteristic of CHS, but patients with CHS tend to maintain normal eating patterns when symptom-free, which further differentiates it from eating disorders.

## 8. Discussion

Cannabis Hyperemesis Syndrome (CHS) represents a growing public health concern, particularly due to the increasing use of cannabis among youth, and the liberalization of cannabis laws. Although the condition is becoming more widely recognized, much of the current evidence on CHS remains of low quality, primarily drawn from case reports and case-series [8]. These limitations underscore the need for further high-quality research, including prospective epidemiological studies, adequately powered studies that enhance methodological rigor and study reliability [57], and randomized controlled trials to establish effective treatment protocols. Despite these challenges, it is clear that CHS has a significant impact on both physical and mental health, and its prevalence is likely to rise as cannabis use becomes more widespread.

A critical issue in managing CHS is the need for greater awareness among clinicians (e.g., emergency doctors, gastroenterologists, neurologists, and psychiatrists). Current medical education and training do not sufficiently address CHS, often leading to delays in diagnosis and inappropriate treatments. Educating healthcare providers about the association between chronic cannabis use and CHS is essential for early intervention and effective management [8,21]. Public health campaigns are also necessary to raise awareness among cannabis users, helping them recognize the potential risks of prolonged cannabis consumption and prompting them to seek care when symptoms arise [45].

The pathophysiology of CHS remains poorly understood, and further basic science research is needed to elucidate the underlying mechanisms. Studies exploring the role of the endocannabinoid system, genetic predispositions, and potential biomarkers could significantly improve our understanding of the condition and aid in developing targeted diagnostic and therapeutic strategies [10,19]. Longitudinal research examining genetic factors and biomarkers may help identify individuals at higher risk for CHS and allow for earlier intervention.

The liberalization of cannabis laws has led to a rise in cannabis consumption, especially among youth, which is likely to contribute to an increase in cases of CHS. This highlights the importance of integrating CHS education into public health initiatives, particularly in school-based health programs and primary care settings. Preventative education and harm reduction strategies should be implemented early, addressing both the psychological and physical aspects of cannabis use. Healthcare providers should approach cessation with sensitivity, offering gradual reduction plans and mental health support to help individuals navigate withdrawal and avoid relapse [20]. For a concise overview of key considerations and potential directions for future research and clinical practice regarding CHS, please refer to the Panel Discussion Box 1.

Box 1**Summary of key recommendations and future priorities for addressing Cannabis Hyperemesis Syndrome (CHS) in youth and** **clinical settings.**
**Panel Discussion Box: Key Considerations for Cannabis Hyperemesis Syndrome (CHS) Moving Forward**
1. Prevention and Early Education for YouthTargeted Awareness Programs: Launch educational campaigns in schools, healthcare settings, and public health platforms to highlight the risks of chronic cannabis use and CHS.Parent and Educator Involvement: Engage parents, teachers, and counselors to help recognize CHS symptoms early.2. Early Recognition and DiagnosisScreening for CHS: Clinicians should screen youth with cyclic vomiting, abdominal pain, and gastrointestinal issues for CHS.Diagnostic Challenges and Misdiagnosis: Avoid confusing CHS with eating disorders like bulimia nervosa by differentiating vomiting from self-induced purging.3. Intervention Strategies for YouthTreatment of Acute Symptoms: Hospital care should focus on rehydration, electrolyte correction, and supportive care, with pharmacological treatments like haloperidol or benzodiazepines as adjuncts. Long-term treatment requires cannabis cessation.Support for Withdrawal Symptoms: Monitor and manage withdrawal symptoms (e.g., anxiety), providing counseling to reduce relapse risks.4. Addressing the Growing Prevalence of CHSImpact of Legalization: As cannabis becomes more accessible, public health messages should highlight the risks of chronic use, especially for youth.Public Health Messaging: Utilize campaigns to educate cannabis users about CHS risks.5. Research Priorities and Future DirectionsPathophysiology of CHS: Urgent need for research into the mechanisms behind CHS.Epidemiological Studies and RCTs: Conduct prospective studies and randomized controlled trials to better understand CHS prevalence, risk factors, and effective treatments.6. Building Clinical and Community SupportTraining for Healthcare Providers: Train providers to recognize CHS and understand its treatment.Integration into Substance Use Treatment: Include CHS in substance use treatment frameworks for a comprehensive approach to care.7. The Role of Policy and AdvocacyAdvocacy for Public Health Policy: Advocate for public health policies prioritizing cannabis education to allocate resources for CHS research, treatment, and prevention.Legislation to Protect Youth: Enforce regulations that limit youth access to cannabis and educate about its long-term risks, including CHS

This narrative review has some limitations. First, much of the existing literature on CHS relies on case reports and observational studies, which may limit the generalizability and reliability of the findings. Second, the pathophysiology of CHS remains poorly understood, with a focus on clinical observations rather than mechanistic research. Lastly, despite efforts to include all relevant studies, the narrative nature of this review means some research may have been overlooked.

## 9. Conclusions

While CHS is increasingly recognized as a significant public health issue, the quality of current research remains limited, and much work remains to be performed to fully understand and address the condition. Enhancing clinician education, expanding research into CHS pathophysiology, and promoting public health awareness are crucial steps in mitigating the impact of this emerging syndrome. Future longitudinal studies and genetic research will be key to developing more precise diagnostic and treatment strategies, ultimately reducing the burden of CHS on patients and the healthcare system.

## Figures and Tables

**Figure 1 ijerph-22-00633-f001:**
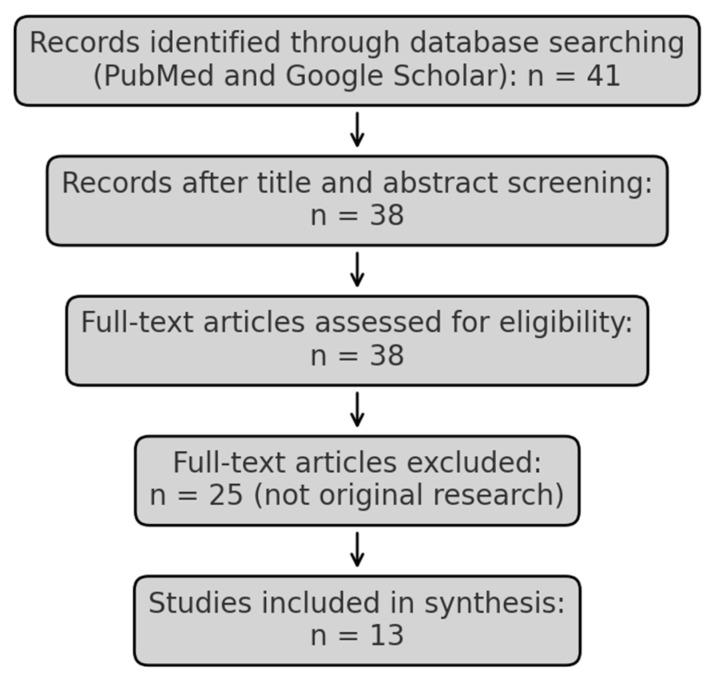
Modified PRISMA flow diagram.

**Table 1 ijerph-22-00633-t001:** Summary of included studies on cannabis hyperemesis syndrome in youth (N = 13).

Author(s), Year	Objective	Study Design	Sample Size	Key Findings
Allen et al., 2004 [27]	To explore the association between chronic cannabis abuse and cyclical vomiting illness in a series of cases in South Australia.	Case series study with follow-up through clinical consultations and urine drug screenings.	9 cases analyzed (19 identified, 5 excluded, 5 lost to follow-up).	In all cases, chronic cannabis use preceded cyclical vomiting. Cessation of cannabis led to symptom resolution in 7 patients. Three patients who did not abstain continued vomiting. 9/10 patients displayed abnormal washing behavior during illness episodes.
Attout et al., 2020 [17]	To describe the characteristics of CHS and its association with cannabis use, focusing on two case studies.	Case report of two patients seeking emergency care for recurrent nausea and vomiting.	2 cases.	CHS is under-recognized and often leads to unnecessary investigations. It is linked to chronic cannabis use, with symptoms of cyclic nausea, vomiting, and compulsive hot bathing. Abstinence from cannabis resolved symptoms in both cases.
Chandra et al., 2019 [15]	To examine trends in cannabis potency over the past decade through the University of Mississippi’s potency monitoring program.	Longitudinal analysis of cannabis samples from the University of Mississippi’s program, using GC/FID method.	18,108 cannabis samples analyzed.	Over the past decade, the mean Δ9-THC concentration in cannabis has nearly doubled, from 8.9% in 2008 to 17.1% in 2017. The Δ9-THC:CBD ratio also rose significantly from 23 to 104. Cannabis is becoming more potent and potentially more harmful in the US and several European countries.
Dezieck et al., 2017 [34]	To evaluate the effectiveness of topical capsaicin in treating CHS.	Case series of 13 patients with CHS treated with topical capsaicin in emergency departments.	13 patients.	All 13 patients experienced symptom relief after topical capsaicin administration, suggesting its potential effectiveness in treating CHS after other treatments failed.
El Sherif et al., 2024 [35]	To describe the clinical presentation and treatment of CHS in a patient following cessation of cannabis use.	Case report of a single patient presenting with CHS symptoms after recent cannabis cessation.	1 patient.	The patient developed CHS symptoms after stopping cannabis use. Treatment with a combination of tramadol, promethazine, and mirtazapine led to full recovery within 10 days.
Figueroa-Rivera et al., 2015 [36]	To present a case of CHS and emphasize its clinical features.	Case report of a single patient with recurrent vomiting and compulsive bathing, strongly suggestive of CHS.	1 patient.	The patient’s recurrent vomiting and compulsive bathing strongly correlated with CHS. The syndrome should be considered in patients with a history of chronic cannabis use and intractable vomiting.
Fleig & Brunkhorst, 2015 [37]	To report a case of CHS with persistent hyperemesis and abnormal bathing behavior.	Case report of a single patient with CHS and abnormal bathing behavior.	1 patient.	The patient exhibited persistent hyperemesis and abnormal bathing behavior as the only relief for nausea. Detoxification led to symptom resolution. This is the first reported case of CHS in Germany.
Lonsdale et al., 2021 [38]	To report a case series of adolescent patients with Cannabis Hyperemesis (CH) and evaluate clinical features, diagnostic criteria, and treatment approaches.	Case series of 34 adolescent patients diagnosed with CH at a single institution over 10 years.	34 patients.	Adolescent patients with CH presented with cyclic vomiting, abdominal pain, and relief from hot showers. No specific antiemetic was found to be effective.
Morris & Fisher, 2014 [39]	Present a case of CHS and review the general issues related to CVS.	Case report.	1 patient (20-year-old female).	CHS diagnosed in a patient with chronic marijuana use; compulsive hot water bathing was reported as part of the syndrome.
Myran et al., 2022 [30]	Examine changes in the number and characteristics of CHS emergency department visits before and after cannabis legalization in Ontario, Canada	Repeated cross-sectional study with interrupted time-series analyses	12,866 ED visits from 8140 individuals	No immediate or gradual change in ED visits for CHS after legalization, but commercialization during the COVID-19 pandemic was associated with a 1.49-fold increase in visits. Increases were higher in women and those aged 19–24 years.
Nourbakhsh et al., 2019 [22]	Examine the potential link between CHS and fatal outcomes in chronic cannabinoid users.	Case report.	3 cases.	The deaths of a 27-year-old female, a 27-year-old male, and a 31-year-old male with a history of CHS are reported. All had cyclical nausea and vomiting, chronic cannabinoid use, and negative lab, radiological, and endoscopic findings. CHS was determined to be the cause of death in two cases. Toxicological analysis revealed tetrahydrocannabinol in postmortem blood.
Ruberto et al., 2021 [40]	Compare the efficacy of haloperidol versus ondansetron for the treatment of CHS.	Randomized controlled trial.	33 subjects.	Haloperidol was superior to ondansetron in reducing abdominal pain and nausea in cannabis users with CHS. Haloperidol also led to less use of rescue antiemetics and a shorter time to ED departure. There were two return visits for acute dystonia, both in the higher-dose haloperidol group.
Young-Wolff et al., 2024 [41]	To explore clinician perspectives on the impact of recreational cannabis legalization (RCL) on adolescent cannabis-related beliefs, behaviors, and health consequences.	Qualitative study using semi-structured interviews and thematic analysis.	32 clinicians.	Clinicians reported increases in adolescent cannabis use, earlier onset, use of high-potency and non-combustible forms, and cannabis-related issues like CHS and psychosis post-RCL. They also noted shifts in social norms, easier access, increased parental use/permissiveness, reduced perceived harm, and decreased court-mandated treatment.
Zhu et al., 2021 [29]	To synthesize qualitative and quantitative data on the diagnosis and effective management of CHS in adolescents.	Systematic review.	21 studies included (from an initial pool of 1334 articles).	CHS diagnostic criteria in adolescents align with adults but may present earlier and more often in females. 21% of adolescent CHS patients had a history of anxiety/depression. Haloperidol and capsaicin may help symptomatically, but cannabis cessation is the only consistently effective treatment.

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
