# Peer review of "Cannabis Hyperemesis Syndrome in Youth: Clinical Insights and Public Health Implications"

_ijerph, 2025, doi:10.3390/ijerph22040633_

Round 1
Reviewer 1 Report
Comments and Suggestions for Authors
This review is focused on Cannabis Hyperemesis Syndrome in youth, including its progression, impact on health, and treatment. In general, I believe this work can make a potentially useful contribution, as CHS is a major medical concern for certain people who use cannabis, but at present, relatively little is known about the condition. However, the authors do not clearly lay out the type of review this is intended to be, nor explain their search methods or the criteria for deciding what to include in the review. This makes it difficult to know how this paper fits into the current literature on CHS. This and several related weaknesses reduce my enthusiasm. Below are some specific suggestions for improving the manuscript.
- In the second paragraph, the authors introduce the idea that increasing THC potency is a concern and then suggest that high potency cannabis is linked to CHS. However, they do not provide literature supporting this link. This is an important point that sets up the need for the review, and it should be backed up by references when this idea is first introduced in the text.
- This is clearly not a systematic review, but even as a narrative review (which seems like the best categorization for this paper) there should be some information provided on the search methods/how the authors decided what literature to include. As it is written, it is somewhat unclear how the authors made these decisions. The authors could also consider including “narrative review” in the title.
- Related to the above point, the authors state that “this article aims to provide a comprehensive overview of CHS in youth”. What makes this review comprehensive and/or how does it fill gaps in the literature? They list the topics they intend to cover, but it would be helpful to know, for example, what this paper will add beyond what is provided in prior reviews (e.g., Zhu et al., 2021, Journal of Adolescent Health)?
- It would be helpful in the introduction section to contextualize why CHS was not documented in the literature before 2004.
- In section 3, very little information is provided on the “recovery” phase from CHS. More detail here would be useful, including information on timeframe, potential challenges, any individual differences, etc.
- In section 4, I recommend mentioning that some individuals may start using cannabis initially to cope with mental health symptoms such as anxiety and depression. While it is understood that CHS symptoms may sometimes cause depression and anxiety, it may also be the case that people who use cannabis had pre-existing anxiety and depression, and the CHS symptoms are not necessarily causing these symptoms but may be exacerbating pre-existing mental health concerns. Section 4 could also benefit from some comment on how/whether the other negative effects of chronic cannabis use on physical and mental health intersect with the effects of CHS specifically.
- This reviewer is unsure what exactly is meant by “Youth on the Edge” in the title. Would recommend a more descriptive/straightforward title.
- The statement on lines 103-106 “Adolescents 103 are known to experience higher levels of stress than older individuals” warrants a reference (the reference provided at the end of that sentence appears to be supporting the second part of the sentence regarding genotype X environment interactions).
- It seems that a major barrier to proper diagnosis of CHS (especially in youth) is that providers often do not (at least initially) know that their patient has been using cannabis. Thus, in addition to their recommendation that providers be trained to screen for CHS, I think it could be useful for the authors to make evidence-based recommendations about how to increase the chances of young people being willing to disclose to medical providers that they use cannabis. Perhaps educating medical providers on non-stigmatizing language to use with people who come to the ED with severe vomiting (e.g., how to phrase questions about drug use)? I did appreciate the suggestion in section 5.3 that medical providers should approach cannabis cessation from a harm reduction perspective, but even this conversation cannot happen if a patient is unwilling to disclose their cannabis use.
Author Response
This review is focused on Cannabis Hyperemesis Syndrome in youth, including its progression, impact on health, and treatment. In general, I believe this work can make a potentially useful contribution, as CHS is a major medical concern for certain people who use cannabis, but at present, relatively little is known about the condition. However, the authors do not clearly lay out the type of review this is intended to be, nor explain their search methods or the criteria for deciding what to include in the review. This makes it difficult to know how this paper fits into the current literature on CHS. This and several related weaknesses reduce my enthusiasm.
Thank you for your kind words. As noted below, we have added a Materials and Methods section to the manuscript and addressed each of your comments below.
Below are some specific suggestions for improving the manuscript.
In the second paragraph, the authors introduce the idea that increasing THC potency is a concern and then suggest that high potency cannabis is linked to CHS. However, they do not provide literature supporting this link. This is an important point that sets up the need for the review, and it should be backed up by references when this idea is first introduced in the text.
That is a fair point. We have added a reference from a systematic review and included the following in the third paragraph: “While definitive causation has not been established, emerging evidence suggests that chronic use of high-potency cannabis may be associated with a heightened risk of developing CHS [8].”
This is clearly not a systematic review, but even as a narrative review (which seems like the best categorization for this paper) there should be some information provided on the search methods/how the authors decided what literature to include. As it is written, it is somewhat unclear how the authors made these decisions. The authors could also consider including “narrative review” in the title.
Thank you for this point. This is indeed a narrative review. We have added a Materials and Methods section to the manuscript, providing a detailed description of the search strategy and literature selection process. We believe this improves the transparency and rigor of the review. Please see our comment below with regard to your other suggestion about the title of the manuscript.
Related to the above point, the authors state that “this article aims to provide a comprehensive overview of CHS in youth”. What makes this review comprehensive and/or how does it fill gaps in the literature? They list the topics they intend to cover, but it would be helpful to know, for example, what this paper will add beyond what is provided in prior reviews (e.g., Zhu et al., 2021, Journal of Adolescent Health)?
We appreciate the reviewer’s comment and agree that clarifying the novelty and comprehensiveness of our review is important. In response, we have revised the introduction to better articulate the unique contribution of this paper:
Given the limited awareness of CHS among both cannabis users and healthcare providers [5,6,26,28,33], this article provides a comprehensive and up-to-date overview of CHS in youth. Building on recent advancements in the literature, it synthesizes the latest evidence on CHS symptoms, diagnostic challenges, and phases of the condition, while also examining its physical and mental health impacts. This review goes further by addressing nutritional considerations, potential misdiagnosis with eating disorders, and the implications of cannabis legalization on youth health. With cannabis use becoming more prevalent and socially accepted, especially among young people, it is critical to raise awareness of CHS and support early intervention, prevention strategies, and clinical education. We hope this expanded scope and recency of evidence will make the paper a useful reference for both clinicians and public health professionals.
It would be helpful in the introduction section to contextualize why CHS was not documented in the literature before 2004.
While this is discussed in more detail later in the manuscript, we have modified the last two sentences of the third paragraph 3 in the Introduction to state “CHS was not documented in the literature until 2004 [27], likely due to limited clinical recognition of the syndrome and a lack of formal case documentation. As a result, many clinicians and cannabis users remain unaware of the condition [28], highlighting the need for increased education and awareness.”
In section 3, very little information is provided on the “recovery” phase from CHS. More detail here would be useful, including information on timeframe, potential challenges, any individual differences, etc.
That is a good point. Like our manuscript, we have noticed that the “recovery” phase often gets less attention in the literature as well. While we previously only had a single line on the recovery phase, we have expanded this and it now reads as follows: In the recovery phase, CHS symptoms fully resolve, marked by a return to normal bathing habits, regular eating patterns, and weight gain [21,26]. This phase typically begins within days to weeks of cannabis cessation [21]. One of the main challenges, however, is maintaining cannabis abstinence; symptom recurrence is common in individuals who resume cannabis use, which can perpetuate the cycle of illness [21,26]. While many patients experience complete symptom resolution with sustained abstinence, others may require behavioral support or addiction counseling to prevent relapse [21].
In section 4, I recommend mentioning that some individuals may start using cannabis initially to cope with mental health symptoms such as anxiety and depression. While it is understood that CHS symptoms may sometimes cause depression and anxiety, it may also be the case that people who use cannabis had pre-existing anxiety and depression, and the CHS symptoms are not necessarily causing these symptoms but may be exacerbating pre-existing mental health concerns. Section 4 could also benefit from some comment on how/whether the other negative effects of chronic cannabis use on physical and mental health intersect with the effects of CHS specifically.
We appreciate these two recommendations and have now added two additional paragraphs to section 5 (formerly section 4) addressing these issues.
This reviewer is unsure what exactly is meant by “Youth on the Edge” in the title. Would recommend a more descriptive/straightforward title.
We have given this comment significant thought. While we thought our title was quite compelling, we have changed our title to better reflect the paper’s focus. Our title has been changed to: Cannabis Hyperemesis Syndrome in Youth: Clinical Insights and Public Health Implications.
The statement on lines 103-106 “Adolescents 103 are known to experience higher levels of stress than older individuals” warrants a reference (the reference provided at the end of that sentence appears to be supporting the second part of the sentence regarding genotype X environment interactions).
Thank you for your comment. We respectfully note that the reference provided (Seabrook & Avison, 2010) supports both components of the sentence. In that paper, we discuss how adolescents experience disproportionately high levels of stress relative to older individuals and how this elevated stress can interact with genetic predispositions to influence health outcomes. To improve clarity, we have revised the sentence slightly to better reflect the connection between the two ideas and their joint support by the cited reference: "Adolescents are known to experience higher levels of stress than older individuals, a developmental reality that can interact with genetic predispositions to increase the likelihood of adverse health outcomes [37]."
It seems that a major barrier to proper diagnosis of CHS (especially in youth) is that providers often do not (at least initially) know that their patient has been using cannabis. Thus, in addition to their recommendation that providers be trained to screen for CHS, I think it could be useful for the authors to make evidence-based recommendations about how to increase the chances of young people being willing to disclose to medical providers that they use cannabis. Perhaps educating medical providers on non-stigmatizing language to use with people who come to the ED with severe vomiting (e.g., how to phrase questions about drug use)? I did appreciate the suggestion in section 5.3 that medical providers should approach cannabis cessation from a harm reduction perspective, but even this conversation cannot happen if a patient is unwilling to disclose their cannabis use.
That is a fair point. I have added a new subsection addressing these very points in Section 6.5.
Reviewer 2 Report
Comments and Suggestions for Authors
Thank you very much for the opportunity to read this manuscript. This study explored the Cannabis Hyperemesis Syndrome (CHS) progression in youth, its impact on health, and assess current treatment strategies. The results offer a deeper understanding of a phenomenon that is increasing with the change in medical and recreational cannabis laws and will be valuable for healthcare professionals and services in developing targeted strategies to support youth. The following are suggestions to further strengthen this paper:
Although, you mention in the abstract that this study is a review, there is no description of the methods used to conduct this review. Describing the methods is important to increase transparency, reproducibility, and comprehension, and reduce bias. I recommend describing the search strategy (databases searched and search terms and keywords), when the search was conducted, inclusion and exclusion criteria, details of the selection process and data extraction, and how the data were synthesized. Is this a narrative review? Even for a narrative review, it is important to describe this information.
Is this line, I recommend including a PRISMA flow diagram.1
I recommend including a table or box listing the articles included in this review and some information about each article (e.g., article/authors, year of publication, study design, number of participants, proportion of males and females, information about cannabis consumption if available (e.g., average of days of use, THC levels), and main results or themes addressed by the article).
It is important to report the limitations of this study.
Reference
Moher D, Shamseer L, Clarke M, et al. Preferred reporting items for systematic review and meta-analysis protocols (PRISMA-P) 2015 statement. Syst Rev 2015;4:1. doi:10.1186/2046-4053-4-1
Author Response
Thank you very much for the opportunity to read this manuscript. This study explored the Cannabis Hyperemesis Syndrome (CHS) progression in youth, its impact on health, and assess current treatment strategies. The results offer a deeper understanding of a phenomenon that is increasing with the change in medical and recreational cannabis laws and will be valuable for healthcare professionals and services in developing targeted strategies to support youth.
Thanks very much for your kind words.
Although, you mention in the abstract that this study is a review, there is no description of the methods used to conduct this review. Describing the methods is important to increase transparency, reproducibility, and comprehension, and reduce bias. I recommend describing the search strategy (databases searched and search terms and keywords), when the search was conducted, inclusion and exclusion criteria, details of the selection process and data extraction, and how the data were synthesized. Is this a narrative review? Even for a narrative review, it is important to describe this information.
We appreciate this comment. This is indeed a narrative review and we have added a Materials and Methods section to the manuscript, providing a detailed description of the search strategy, when the search was conducted, and the literature selection process.
Is this line, I recommend including a PRISMA flow diagram.
Reference
Moher D, Shamseer L, Clarke M, et al. Preferred reporting items for systematic review and meta-analysis protocols (PRISMA-P) 2015 statement. Syst Rev 2015;4:1. doi:10.1186/2046-4053-4-1
Thank you for your suggestion regarding the inclusion of a PRISMA flow diagram. As this article is a narrative review, we believe that a PRISMA flow diagram is not necessary because our selection process does not follow the formal PRISMA guidelines of a systematic review. In our review, we conducted an iterative search process and included studies based on thematic relevance. The themes were determined in advance, and the search strategy aimed to ensure the most relevant literature was included. However, we appreciate the value of clarity in the presentation of our methodology. As a result, we have provided a detailed description of our search strategy, inclusion/exclusion criteria, and thematic focus within the Materials and Methods section to maintain transparency.
I recommend including a table or box listing the articles included in this review and some information about each article (e.g., article/authors, year of publication, study design, number of participants, proportion of males and females, information about cannabis consumption if available (e.g., average of days of use, THC levels), and main results or themes addressed by the article).
Thank you for this suggestion. While we understand the value of summarizing each article, we believe that a more valuable contribution to this field is our Panel Discussion Box that synthesizes key considerations for CHS moving forward. This section, which includes preventive strategies, early recognition, intervention, public health messaging, and research priorities, offers actionable insights for clinicians, researchers, and policymakers to address the growing challenges associated with CHS. The Panel Discussion Box also serves to guide further research, clinical interventions, and public health efforts in a way that aligns with the objectives of this review. Additionally, we note that the other reviewer did not request such a table, so we hope that the Panel Discussion Box will suffice as we feel it has more value moving forward.
It is important to report the limitations of this study.
Thank you for this comment. We had previously written that “Although the condition is becoming more widely recognized, much of the current evidence on CHS remains of low quality, primarily drawn from case reports and case-series [8]. These limitations underscore the need for further high-quality research, including prospective epidemiological studies, adequately powered studies that enhance methodological rigor and study reliability [57], and randomized controlled trials to establish effective treatment protocols." However, we agree that more explicit limitations should be acknowledged, so we added the following paragraph write before the Conclusions subsection:
“This narrative review has some limitations. First, much of the existing literature on CHS relies on case reports and observational studies, which may limit the generalizability and reliability of the findings. Second, the pathophysiology of CHS remains poorly understood, with a focus on clinical observations rather than mechanistic research. Lastly, despite efforts to include all relevant studies, the narrative nature of this review means some research may have been overlooked.”
Round 2
Reviewer 1 Report
Comments and Suggestions for Authors
The authors have sufficiently addressed my comments. I have no further suggestions.
Author Response
The authors have sufficiently addressed my comments. I have no further suggestions.
Thank you.
Reviewer 2 Report
Comments and Suggestions for Authors
Thank you for the opportunity to read this manuscript. This study explored the Cannabis Hyperemesis Syndrome (CHS) progression in youth, its impact on health, and assess current treatment strategies. The revisions enhanced clarity and strengthened the implications of the article. However, the following are suggestions to further strengthen this paper:
Although this is a narrative review, including PRISMA flow diagram(1) would give the reader a clear view of how many articles were found in the search, as well as the number of articles included and excluded.
I also recommend including a table or box listing the articles included in this review and some information about each article (e.g., article/authors, year of publication, study design, number of participants, proportion of males and females, information about cannabis consumption if available (e.g., average days of use, THC levels), and main results or themes addressed by the article). Including a visual representation will help readers easily comprehend the results and gain a broader understanding of the types of studies investigating CHS.
Reference
(1) Moher D, Shamseer L, Clarke M, et al. Preferred reporting items for systematic review and meta-analysis protocols (PRISMA-P) 2015 statement. Syst Rev 2015;4:1. doi:10.1186/2046-4053-4-1
Author Response
Although this is a narrative review, including PRISMA flow diagram(1) would give the reader a clear view of how many articles were found in the search, as well as the number of articles included and excluded.
I also recommend including a table or box listing the articles included in this review and some information about each article (e.g., article/authors, year of publication, study design, number of participants, proportion of males and females, information about cannabis consumption if available (e.g., average days of use, THC levels), and main results or themes addressed by the article). Including a visual representation will help readers easily comprehend the results and gain a broader understanding of the types of studies investigating CHS.
Reference
(1) Moher D, Shamseer L, Clarke M, et al. Preferred reporting items for systematic review and meta-analysis protocols (PRISMA-P) 2015 statement. Syst Rev 2015;4:1. doi:10.1186/2046-4053-4-1
Thank you for these recommendations. We have added a comprehensive summary table (Table 1) that outlines all 41 articles included in this narrative review. This table includes the author(s) and year of publication, objective, study design, sample size, and key findings. We believe this table will provide readers with a clearer understanding of the types of studies included in this review and their findings. Moreover, we added the following in the last paragraph of the Methods section: “Table 1 provides an overview of the 41 articles included in this review, spanning publications from 2004 to 2024. These articles consisted of 21 narrative reviews, 9 case-based studies (6 case reports and 3 case series), 4 expert commentaries or opinion pieces, 2 systematic reviews, 1 scoping review, 1 randomized controlled trial, 1 longitudinal analysis, 1 repeated cross-sectional study, and 1 qualitative study.”
We appreciate your suggestion regarding the inclusion of a PRISMA flow diagram. However, as this is a narrative review, rather than a systematic review or meta-analysis, we did not employ the PRISMA guidelines for our methodology. Nonetheless, in addition to our description of the article selection process in the Methods section, we hope that our new summary table (Table 1) will offer sufficient clarity regarding the studies included in this review.